# Preventive Potential of Dipeptide Enterocin A/P on Rabbit Health and Its Effect on Growth, Microbiota, and Immune Response

**DOI:** 10.3390/ani12091108

**Published:** 2022-04-26

**Authors:** Monika Pogány Simonová, Ľubica Chrastinová, Jana Ščerbová, Valentína Focková, Iveta Plachá, Zuzana Formelová, Mária Chrenková, Andrea Lauková

**Affiliations:** 1Centre of Biosciences of the Slovak Academy of Sciences, Institute of Animal Physiology, Šoltésovej 4-6, 040 01 Kosice, Slovakia; scerbova@saske.sk (J.Š.); fockova@saske.sk (V.F.); placha@saske.sk (I.P.); laukova@saske.sk (A.L.); 2National Agricultural and Food Centre, Hlohovecká 2, 951 41 Lužianky, Slovakia; lubica.chrastinova@nppc.sk (Ľ.C.); zuzana.formelova@nppc.sk (Z.F.); maria.chrenkova@nppc.sk (M.C.)

**Keywords:** enterocin, rabbit, health, immunity, microbiota, prevention

## Abstract

**Simple Summary:**

Rabbits are animals sensitive to alimentary disturbances and various spoilage agents, mostly during the weaning period. For this reason, the use of natural feed additives has become an area of research in rabbit nutrition, mainly with a focus on prevention. The “in vivo” administration of bacteriocins/enterocins shows an increasing potential in the prevention/treatment of animals’ diseases. Therefore, our study focused on the preventive potential of the dipeptide enterocin (Ent) A/P against the methicillin-resistant (MR) *Staphylococcus epidermidis* SE P3/Tr2a strain in rabbit model, determining its effect on the growth performance, phagocytic activity, secretory (s) IgA, and gut microbial composition of rabbits. Ent A/P increased the weight gain of rabbits and its antibacterial effect showed a tendency to stabilize and improve gut microbiota due to reduction of MR staphylococci, total bacteria, and coliforms. The immune-stimulatory effect of Ent A/P was noted due to increased phagocytic activity. Achieved results showed the great potential of Ent A/P application as a feed additive in rabbit nutrition to improve the health and productivity of animals.

**Abstract:**

The present study investigated the effect of the dipeptide enterocin (Ent) A/P on growth, immune response, and intestinal microbiota in rabbits. Eighty-eight rabbits (aged five weeks, M91 meat line, both sexes) were divided into three experimental groups: E (Ent A/P; 50 µL/animal/day for 14 days; between 0–14 days); S (methicillin-resistant *Staphylococcus epidermidis* SE P3/Tr2a strain; 500 µL/animal/day for 7 days starting at day 14 to day 21); and E + S (Ent A/P between 0–14 days and SE P3/Tr2a strain between 14–21 days) groups, and the control group (C). The additives were administered in drinking water. Administration of Ent A/P lead to an increase in weight gain, reduction of feed conversion; phagocytic activity was stimulated and gut microbiota were optimized due to reduction of coliforms, total bacterial count, and methicillin-resistant staphylococci. Good health and increased weight gain also showed that methicillin-resistant *S. epidermidis* SE P3/Tr2a strain did not have any pathogenic effect on rabbits’ health status.

## 1. Introduction

The routine and indiscriminate use of conventional antibiotics in agriculture presents a growing threat, leading to an increase in drug-resistant bacteria in animals and their transmission to humans. Current research is focusing on alternative antimicrobial compounds for use in animal production and veterinary medicine. Bacteriocins are antimicrobial proteins with a broad antibacterial spectrum produced by Gram-positive and Gram-negative bacteria, mostly by lactic acid bacteria (LAB), including enterococci (producing bacteriocins named mostly enterocins; [1]), currently indicated for use in food preservation, but they also have great potential in the prevention and treatment of animals’ diseases. Bacteriocins/enterocins have a low tendency to develop resistance compared to conventional antibiotics and they are characterized by biosafety and several bioactive roles–they possess antimicrobial, anticancer, antioxidant, and immunomodulatory effects [2,3,4,5]. Therefore, bacteriocins could be a promising alternative as feed additives in livestock farms to improve the health and productivity of food animals. Regarding their success to eliminate multi-drug resistant (MDR) bacteria, they can also be applied in several areas of veterinary medicine with preventive effects and/or to treat bacterial infections [4,6,7].

MDR bacteria represent a public health problem worldwide, with the focus mainly on vancomycin-resistant enterococci, carbapenem-resistant enterobacteria, and pseudomonads. There is also a focus on methicillin-resistant *Staphylococcus aureus* (MRSA) and other coagulase-positive (MRCoPS) and coagulase-negative staphylococci (MRCoNS) [8]. While MRSA in human medicine is detected mostly as hospital-acquired MRSA (HA-MRSA), in veterinary medicine methicillin-resistant staphylococci (MRS) are causative agents of cows’ mastitis, chickens’ lameness, pigs’ exudative epidermitis, canine pyoderma, skin lesions in dogs and cats, and abscesses, mastitis, and septicemia in rabbits [9,10]. Because of a great diversity of MRS host species–pets, food, and wild animals [11]–a special attention is required regarding MRS’ co-resistance to other antibiotics, influencing the antimicrobial therapeutic effect and offering a source of resistance genes for other staphylococcal species. For this reason, new antimicrobial agents directed against antibiotic-resistant bacteria, such as bacteriocins, are requested. In vitro bacteriocins testing against MRS, MRSA, and multi-drug resistant staphylococci (MDRS) showed promising results, including their antimicrobial and antibiofilm effect [12,13,14], confirmed also in several animal models, mostly in the murine model [3]. While the anti-MRSA resp. anti-MRS activity of bacteriocins was presented in a number of papers, mostly during lantibiotics’ testing, the effect of enterocins on MRS isolates are documented only in some studies [15,16,17]. Moreover, most of the mentioned in vivo experiments present the therapeutic effect of bacteriocins tested in animal models [3], but the prevention of pathogenic and resistant bacteria multiplication using bacteriocins as feed additives is also important.

Therefore, the aim of this study was to test the preventive effect of a dipeptide enterocin (Ent) A/P (previously named Ent EK13, produced by *Enterococcus faecium* EK13 strain deponed to the Czech Collection of Microorganisms in Brno, Czech Republic with no. CCM7419 [18]) against methicillin-resistant *Staphylococcus epidermidis* SE P3/Tr2a strain in a rabbit model. The effect on growth performance, immune response, *Eimeria* spp. oocysts’ occurrence, and the intestinal microbiota composition of rabbits was controlled.

## 2. Materials and Methods

### 2.1. Animals Care and Use

The experiment was performed in cooperation with our colleagues at the experimental rabbit facility of the National Agricultural and Food Centre, Research Institute for Animal Production, Nitra, Slovakia. All applicable international, national and/or institutional guidelines for the care and use of animals were followed appropriately, and all experimental procedures were approved by the Institutional Ethic Committee (permission code: SK CH 17016 and SK U 18016) and the State Veterinary and Food Administration of the Slovak Republic (4047/16-221).

### 2.2. Experiment Schedule, and Diet

A total of 88 post-weaned hybrid rabbits, (meat lines M91 and P91, weaned at the age of 35 days, both sexes, equal male to female ratio per treatment), were used in this experiment. Rabbits were divided into three experimental groups (E, S, E + S) and one control group (CG), with 22 animals in each group. The average live weight of rabbits at the start of the experiment was 1091.8 g ± 149.3. The rabbits were kept in standard metal cages (61 × 34 × 33 cm, Kovobel company, Domažlice, Czech Republic), two animals per cage. A cycle of 16 h of light and 8 h of dark was used throughout the experiment. Temperature (20 ± 4 °C) and humidity (70 ± 5%) were maintained throughout the experiment by heating and ventilation systems, and data were recorded continuously with a digital thermograph positioned at the same level as the cages. Animals were fed a commercial pelleted diet for growing rabbits (KV, Tekro-Nitra, Ltd., Nitra, Slovakia; Table 1) during the whole experiment with access to water *ad libitum*. The ingredients and chemical composition of this diet are presented in Table 1. The chemical analyses were conducted according to AOAC [19] and Van Soest et al. [20]. The animals in group E were administered Ent A/P (prepared according to Mareková et al. [21]), a dose of 50 μL/animal/day, with activity of 25,600 AU/mL during the first 14 days of the treatment period (between days 0/1 and 14), to control the preventive effect of tested enterocin A/P (Figure 1). The activity of Ent A/P was tested by the agar spot test according to De Vuyst et al. [22] against the principal indicator strain *E. avium* EA5 (isolated from the feces of piglets, our laboratory). Rabbits in group S received only the methicillin-resistant *Staphylococcus epidermidis* SE P3/Tr2a strain (1.0 × 10^5^ CFU/mL; Pogány Simonová et al. [17]) in their drinking water at a dose of 500μL/animal/day for 7 days, between 14 and 21 days. The strain was marked by rifampicin to differentiate it from the total staphylococci and prepared as described previously by Strompfová et al. [23]. Rabbits in the E + S group firstly consumed the Ent A/P for 14 days (between 0–14 days) and after it the SE P3/Tr2a strain was applied to animals for 7 days, starting at day 14 (when the Ent A/P application was ceased), and finishing at day 21 of the trial. Based on our previous experiments, that these additives can be dissolved in distilled water [21], the additives were applied firstly to 100 mL of drinking water in all cages to ensure that rabbits received the applied additives, and after consuming this volume the rabbits had access to water ad libitum. Control rabbits (group C) had the same conditions, but without additives being applied to their drinking water, and they were fed a commercial diet. Drinking water was provided through nipple drinkers. The experiment lasted for 21 days. 

### 2.3. Growth Performance

Body weight (BW) and feed consumption were measured every week during the experiment; average daily weight gain (ADWG) and feed conversion ratio (FCR) were calculated mathematically. Mortality was recorded daily throughout the whole experiment.

### 2.4. Phagocytic Activity in Blood

For phagocytic activity (PA), blood (*n* = 8) was sampled from the marginal ear vein (*Vena auricularis*) into Eppendorf tubes containing micro-spheric hydrophilic (MSH) particles and heparin at days 0, 14, and 21. Briefly, 50 µL of MSH particle suspension (ARTIM, Prague, Czech Republic) was mixed with 100 µL of blood in an Eppendorf-type test tube and incubated at 37 °C for 1 h. Blood smears were then prepared and stained in accordance with May–Grünwald and Giemsa–Romanowski, and the direct microscopic counting procedure, calculating the number of white cells containing at least three engulfed particles per 100 white cells (monocytes/granulocytes), was used for PA analysis [24].

### 2.5. Immunoglobulin A in Small Intestinal Wall

Eight rabbits in each group were killed at 21 days of the trial (56 d of age) using electronarcosis (50 Hz, 0.3 A/rabbit for 5 s), immediately hung by the hind legs on the processing line and quickly bled by cutting the jugular veins and the carotid arteries. Samples (*n* = 8) of the small intestine were collected for IgA analyses, and the appendix and cecum contents for microbial analysis.

The concentration of immunoglobulin A (IgA) in the intestinal wall was measured using the competitive inhibition enzyme immunoassay technique (Rabbit Immunoglobulin A, IgA ELISA kit, Cusabio, Houston, TX, USA). Samples of intestinal wall were prepared according to Nikawa et al. [25] and analyzed using the Multireader Synergy HTX (Biotek, Shoreline, WA, USA), at the wavelength 450 nm according to the manufacturer of Cusabio kit.

### 2.6. Microbial Analyses and Eimeria spp. Oocysts Detection

Freshly-voided feces were collected using nets mounted under the cages, five nets under 11 cages. Because there were two animals housed in each cage, and in some places the feces were mixed, we decided to collect mixed samples, one mixed sample per net, i.e., five mixture samples per group. Feces were sampled at day 0/1 (at the start of the experiment and Ent A/P application; 10 mixture samples from all rabbits, respectively, from all groups together–initial microbial background, at day 14 (2 weeks after the start of the experiment; the end of Ent A/P and the start of the *S. epidermidis* SE P3/Tr2a strain application; five mixture samples from each group) and at day 21 (56 days of age; the end of SE P3/Tr2a strain application). For microbial testing, samples of feces, cecum, and appendix contents (1 g) were treated using the standard microbiological dilution method (International Organization for Standardization (ISO)). The appropriate dilutions in Ringer solution (pH 7.0; Oxoid Ltd., Basingstoke, Hampshire, England) were plated onto the following media: M-Enterococus Agar (NF-V04503, Difco Laboratories, Detroit, MI, USA) for enterococci, M17 Agar (Difco) for streptococci, MacConkey agar (ISO 7402, Oxoid) for coliforms, Mannitol Salt Agar for coagulase-negative staphylococci (CoNS, ISO 6888), Plate Count Agar (Biomark Laboratories, Pune, India) for total bacterial growth, Oxacillin Resistance Screening Agar (Oxoid) to confirm methicillin-resistant staphylococci. Brain-Heart Infusion agar (Difco) enriched with rifampicin (Rifasynt, Medochemie Ltd., Limassol, Cyprus) was used to determine *S. epidermidis* P3/Tr2a. Bacteria were cultivated at 30 °C and/or 37 °C for 24–48 h depending on the bacterial genera and their counts were expressed in log 10 of colony forming units per gram (log 10 CFU/g ± SD). Randomly picked up representatives of selected bacterial groups were confirmed by MALDI-TOF identification system (Bruker Daltonics, Billerica, MA, USA).

Fecal samples were stored at 4 °C and examined for the *Eimeria* spp. oocysts by the flotation technique, according to McMaster [26]. Oocysts were not differentiated at the species level but designated as *Eimeria* spp.; they were counted microscopically, and the intensity of infection was expressed as oocysts per gram of feces (OPG).

### 2.7. Statistical Analysis

Treatment effects on the growth parameters, microbiota in fecal samples, and phagocytic activity (PA) were analyzed using two-way analysis of variance (ANOVA), followed by Bonferroni post-hoc test for pair-wise comparisons, where appropriate. Fixed effects for the model included period and treatment, and the interaction between them. Statistical analysis of microbiota from cecal and appendix samples, feed conversion and secretory IgA was performed with one-way analysis of variance (ANOVA), followed by Tukey’s post hoc test or pair-wise comparisons, where appropriate. The statistical model included the time, treatment effects and their interaction. When the interaction was significant, Fisher’s Least Significant Difference test (Fisher’s LSD) was applied post hoc to determine significant differences among the means. Statistical analysis of the applied SE P3/Tr2a strain in fecal, cecal and appendix samples was performed using unpaired Student *t*-test. All statistical analyses were performed by the GraphPad Prism statistical software (GraphPad Prism version 9.3.1., GraphPad Software, San Diego, CA, USA). Differences between the mean values of the different dietary treatments were considered statistically significant at *p* < 0.05. Data are expressed as means and standard deviations of the mean (SD).

## 3. Results

### 3.1. Growth Performance of Rabbits

The animals were in good health throughout the experiment. Mortality was noted in groups S, E + S, and C (Table 2). Higher BW and ADWG were recorded in all experimental groups during additives’ application compared to the control data (E: by 20.5%; S: by 17.1%; E + S: by 25.6%; Table 2). The highest BW and ADWG (E; *p* < 0.0001) and the lowest FCR (E: by 10.2%) were noted during Ent A/P application. Lower FCR (by 1.5%) was also noted in the E + S group, compared to C.

### 3.2. Immunoglobulin A in Small Intestinal Wall and Phagocytic Activity in Blood

The time and treatment effect was noted on the PA in this study (*p* < 0.0001; Table 2). Ent A/P addition increased the PA values compared to C (day 14; E, E + S vs. C: *p* < 0.001); higher PA was noted one week after its withdrawal only in group E (day 21; E vs. E + S, C: *p* < 0.001). The SE P3/Tr2a strain application to rabbits decreased PA in the E + S group after 2 weeks of Ent A/P addition (day 21; E + S vs. E, S and C: *p* < 0.001), but, on the contrary, the PA value was elevated in group S, without Ent A/P preventive application (S vs. E, E + S, C: *p* < 0.001). The interaction effect was noted at day 14 (E vs. S, C: *p* < 0.0001; E + S vs. E, C: *p* < 0.0001), and at day 2 (E vs. S, E + S, C: *p* < 0.0001; S vs. E + S, C: *p* < 0.0001; E + S vs. C: *p* < 0.01).

IgA levels were similar among the experimental groups, except the E + S group, showing the highest IgA value (*p* = 0.0220; Table 2).

### 3.3. Microbial Population

In feces, most bacteria were influenced by time, treatment, and their interaction (except amylolytic streptococci; Table 3). At day 14, lower counts of all tested bacterial groups, except MRS, were recorded compared to the initial data (day 0/1) and the highest count of coliforms, amylolytic streptococci, staphylococci, MRS, and total bacteria was noted in the S group on this day. At the end of SE P3/Tr2a strain application in the E + S group (day 21), a significant reduction of enterococci, coliforms, MRS (E + S vs. E, S, C: *p* < 0.001), amylolytic streptococci, and total bacterial count (E + S vs. E, S: *p* < 0.001) was detected. The time and treatment interaction was noted on coliforms already at day 14 (S vs. E: *p* < 0.0001; S vs. E + S: *p* < 0.001; S vs. C: *p* < 0.01). At the end of the experiment, enterococci (E + S vs. E, C: *p*< 0.0001; E + S vs. S: *p* < 0.001), staphylococci, and total bacteria (E vs. E + S: *p* < 0.05; S vs. E + S: *p* < 0.001; S vs. C: *p* < 0.01), MRS (E vs. S: *p* < 0.01; E + S vs. E, C: *p* < 0.0001; E + S vs. S: *p* < 0.001), and coliforms (E vs. E + S: *p* < 0.05; S vs. E + S: *p* < 0.01) were influenced by the time and treatment interaction.

The SE P3/Tr2a strain was able to colonize the digestive tract of rabbits after its one-week application, reaching counts in the range 1.20–1.66 log cycle in the feces (Table 3) and also in the range 0.90–1.00 log cycle in the cecum and appendix (Table 4).

Most of the tested bacteria were enumerated in lower counts in the cecum and appendix (except cecal MRS), than in fecal samples. In the cecum, counts of enterococci and coliforms were reduced significantly (*p* < 0.001, Table 4) compared with C, while MRS and total bacteria were increased in the E + S group (E + S vs. E, C: *p* < 0.001).

Fecal samples in all groups (experimental and control) were *Eimeria* spp. oocytes absent.

## 4. Discussion

Good health status, higher BW, and ADWG of rabbits recorded in all experimental groups during additives’ application indicated the beneficial effect of Ent A/P on the one hand, and on the other hand no negative impact of the SE P3/Tr2a strain on animals’ growth was noted. However, reduced FCR in the E and E + S groups reconfirms the positive influence of Ent A/P on rabbits’ growth performance, which can be explained by better feed intake and consumption; this finding is important from the economic point of view (less feed for faster growth). Faster growth and higher weight gain in rabbits were described in many studies during probiotics and their metabolites’ application [27,28,29,30]. The beneficial effect of bacteriocins on the ADWG and FCR in rabbit experiments was repeatedly confirmed in this study, according to previous results [30,31,32,33]. The preventive effect of Ent A/P before SE P3Tr2a strain application to rabbits is reflected in the ADWG elevation and the FCR reduction compared to data from animals receiving only the SE P3/Tr2a strain, without enterocin treatment.

In addition to the positive impact of beneficial strains and/or their bacteriocins on animals’ growth, their immunostimulatory effect is also important. These bioactive compounds can enhance immunity in several ways, including intestinal microbiota balance and the ability to modulate the host’s innate and specific immune response. In general, stimulation of non-specific immunity is demonstrated by increased phagocytosis and modified cytokine production, while elevated immunoglobulins reflect the specific means of stimulation [34]. However, whilst several works have been published on the immune indices in healthy rabbits, existing knowledge on probiotics and bacteriocins’ effect on phagocytic activity (PA) is still limited and needs to be expanded [35,36,37]. A significant increase of PA after 14 days of Ent A/P administration showed the non-specific immunity enhancement. Higher PA values noted one week after its cessation repeatedly confirmed the prolonged immuno-stimulative effect of Ent A/P, as it was previously described during several bacteriocins/enterocins treatment in rabbits [30,38]. The long term immuno-moderating effect of enterocins can be explained either by the maintenance of intestinal health, microbiota, and immunity via supporting the gut-associated lymphoid tissue (GALT), or by the adopting of animals on them. This finding also confirms the use of bacteriocins with a preventive effect in animals, compared to beneficial bacteriocin-producing strains, possessing immunoactivity already during their application and demonstrating more of the therapeutic effect of probiotics. The protective effect of Ent A/P was noted in the E + S group, as no increase in PA was noted after the SE P3/Tr2a strain application, while the strain itself without Ent A/P stimulated the rabbit immune system.

Several authors presented improved cell-mediated immunity [37] and an increased number of mast cells in the cecum, as well as higher IgM and IgG levels in serum [39] after probiotic administration to rabbits. It is well known that natural feed additives can stimulate immunity via enhancing rabbits’ gut health. The predominant isotype of the mucosal immune system is the secretory IgA (sIgA), which can keep the mucosal immunity balance between commensals’ microbiota and pathogens’ defenses on the mucosal surface [40]. SIgA is synthesized by plasma cells in the lamina propria and translocated through intestinal epithelial cells, serving as the first line of defense to protect the intestinal epithelium from enteric toxins and pathogenic microorganisms by immune exclusion. Liu et al. [41] presented maintained intestinal barrier function and increased sIgA production after *Clostridium butyricum* probiotic administration to weaning rabbits, similarly to Plachá et al. [42] who detected a higher amount of IgA in the intestinal wall after thymol application to rabbits. In contrast to Chen et al. [43], who reported increased bile IgA in bacteriocin-treated broiler chicks, our results showed only slight (not significant) sIgA secretion after Ent A/P and SE P3/Tr2a strain application separately. On the other hand, SE P3/Tr2a application after preventive administration of Ent A/P significantly elevated the sIgA concentration. From this result, we assume that the higher amount of IgA detected in the intestinal wall is the result of the primary immuno-stimulation that is evoked by Ent A/P, and the secondary induced by methicillin-resistant SE P3/Tr2a strain as a possible agent. Given these initial results regarding the enterocin-protective effect, further studies on IgA secretion and mucosal immunity in rabbits are needed.

In addition to healthy intestinal immunity, a stable gut microbiome also contributes to the formation of animals’ gut health. This microbial and immunological stability is often negatively influenced by several exogenous factors (stress, dietary and climate changes, etc.), but also can be improved mostly by natural feed additives, via strengthening the intestinal barriers, optimizing microbial balance, and supporting enzymatic activities and nutrient absorption in the gut. The in vivo antimicrobial effect of bacteriocins/enterocins in rabbits’ gastrointestinal tract has been presented in several works [29,30,44,45,46], when mostly the counts of enterococci, clostridia, coliforms, and staphylococci were reduced. In rabbits, colibacillosis and clostridiosis are the main infections, particularly around the weaning period, when the animals are the most susceptible to these bacteria. Staphylococci as a part of the skin and mucosal barriers, and digestive tract as well, are also often detected as the main causative agents of skin lesions and abscesses and/or as associated pathogens in other multifactorial/multiorgan infections. Several works present the anti-staphylococcal activity of bacteriocins, including also MRS isolates [7,15,17]. While enterococci, coliforms and total bacteria were reduced at the end of Ent A/P application (14 days of the trial) compared to their initial data (day 0/1), balanced counts of tested bacteria within the experimental and control groups indicated no significant antibacterial activity of Ent A/P, including its anti-staphylococcal activity. These findings are contradictory to previously observed results during in vitro and in vivo enterocins’ administration [15,30,31,32,33,45]. At day 21, at the end of SE P3/Tr2a strain application to rabbits, enterococci and coliforms significantly decreased in the E + S group, while staphylococci were noted in the highest counts; this fact may also indicate a possible competitive effect between the microbiota established in the gut and the applied SE P3/Tr2a strain. For this reason, we also expected a multiplication of the MRS bacteria, but, surprisingly, the lowest MRS counts were noted in the E + S group. On the other hand, this significant MRS reduction in the E + S group compared to negative (group C) and positive (group S) control at day 21 (after one week SE P3/Tr2a strain application) indicated the anti-MRS effect of Ent A/P, previously confirmed under enterocins in vitro testing against methicillin-resistant staphylococcal isolates [17]. The inhibitory effect on enterococci and coliforms was repeatedly confirmed in cecal samples of rabbits, similar to previous studies [30,31,32,33,45]. A significant increase of MRS and the total bacteria in the cecum, and their higher counts compared to fecal samples at day 21, can be explained as result of re-colonization with SE P3/Tr2a strain due to cecotrophy in the rabbits. To expand these results and already known facts concerning rabbits’ health, immunity, and microbial balance by additives’ supplementation is necessary through further investigations.

Coccidiosis is the most frequent parasitic disease in rabbit farms, affecting all ages, especially young, weaned rabbits, and causing high morbidity and mortality rates [46]. The oocysts are always present in the intestines of rabbits, and they cannot be eliminated even by the use of coccidiostat. Because the EU has banned antibiotics as feed additives and growth promoters, they are replaced with alternative anticoccidials, including prebiotics and probiotics, based on their bactericidal and/or bacteriostatic activities. Although *Eimeria* spp. oocytes were not detected in the feces of rabbits, the anticoccidial effect of Ent A/P can be expected, similarly to the anticoccidial effect of Ent 7420 noted in rabbits [30]; however, further studies are needed.

## 5. Conclusions

Good health and increased weight gain reflect the beneficial effects of Ent A/P on the growth performance of rabbits. These results also showed that the methicillin-resistant S. epidermidis SE P3/Tr2a strain did not have any pathogenic effect on rabbits’ health status. The preventive effect of Ent A/P was recorded due to improved zootechnical parameters (elevated ADWG, reduced FCR), stimulated non-specific immunity (higher PA), and stabilized intestinal microbial environment (stable sIgA, reduced counts of coliforms, staphylococci, amylolytic streptococci, and total bacterial count as well) of rabbits. Reduced MRS and total bacteria after one week of SE P3/Tr2a strain application also point to the antibacterial effect of dipeptide Ent A/P and its anti-MRS potential. This study has an impact on basic research, confirming the in vivo anti-MRS activity of enterocin and also helping to spread knowledge regarding the interaction between intestinal microbiota and immunity.

## Figures and Tables

**Figure 1 animals-12-01108-f001:**
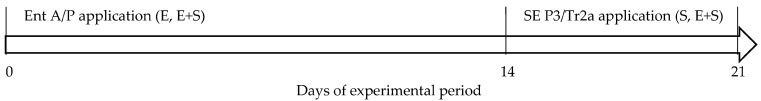
Scheme of the Ent A/P and S. epidermidis SE P3/Tr2a strain application.

**Table 1 animals-12-01108-t001:** Nutrient content of commercial granulated diet for growing rabbits.

Nutrient Content	g·kg^−1^ in Original Feed	g·kg^−1^ in Dry Matter
Dry matter	886.65	1000.00
Crude protein	155.35	174.94
Crude fiber	132.37	149.29
Crude fat	20.30	22.89
Ash	90.08	101.60
Starch	238.71	269.22
Acid detergent fiber	151.69	171.08
Neutral detergent fiber	295.10	332.83
Calcium	15.90	17.94
Phosphorus	4.89	5.51
Magnesium	2.57	2.90
Sodium potassium	1.21	1.36
Iron	564.70 *	636.88 *
Zinc	97.77 *	110.27 *
Copper	20.50 *	23.12 *
Metabolizable energy (MJ.kg−1)	11.16	11.02

* mg·kg^−1^ feed.

**Table 2 animals-12-01108-t002:** Growth performance, phagocytic activity, and secretory IgA of rabbits.

Tested Parameters	Day	E	S	E + S	C	*p*-Value		
						**Time**	**Treatment**	**Time × Treatment**
Body weight (g)	0	1153.0 ± 178.2 ^a^	1140.2 ± 149.3 ^ab^	1060.9 ± 150.8 ^ab^	1012.9 ± 118.8 ^b^			
	14	1735.0 ± 260.0 ^a^	1734.5 ± 152.3 ^ab^	1691.5 ± 213.1 ^ab^	1487.8 ± 203.9 ^b^	**<0.0001**	**<0.0001**	0.2982
	21	2052.7 ± 266.1 ^a^	2014.8 ± 186.1 ^ab^	1999.0 ± 212.2 ^ab^	1759.7 ± 218.6 ^b^			
Average daily weight gain (ADWG; g/day/rabbit)	0–14	41.58 ± 5.85 ^a^	42.45 ± 8.67 ^ab^	45.04 ± 9.54 ^ab^	33.92 ± 8.53 ^b^	0.2940	**<0.0001**	0.0997
	14–21	45.39 ± 8.13 ^a^	40.04 ± 8.82 ^ab^	43.93 ± 7.80 ^ab^	38.85 ± 10.61 ^b^			
Mortality (rabbit/group)	0–21	0	1	2	1			
Feed conversion (g/g)	0–21	2.37 ± 0.44	2.78 ± 0.92	2.60 ± 0.67	2.64 ± 0.20	0.1493		
Phagocytic activity (PA; %)	0	63.75 ± 3.33	63.75 ± 3.33	63.75 ± 3.33	63.75 ± 3.33			
	14	69.63 ± 2.13 ^Aa^	61.63 ± 2.77 ^Bb^	71.00 ± 2.14 ^Aa^	59.63 ± 3.66 ^Bb^	**0.0097**	**<0.0001**	**<0.0001**
	21	70.25 ± 1.39 ^Aa^	81.13 ± 1.60 ^Bb^	53.88 ± 3.60 ^Cc^	58.25 ± 2.82 ^Dd^			
Secretory IgA (μg/g)	21	9.817 ± 0.765 ^a^	9.929 ± 0.635 ^a^	17.240 ± 3.694 ^b^	9.705 ± 0.761 ^a^	**0.0220**		

E–Ent A/P application between 0–14 days; S–SE P3/Tr2a application between 14–21 days; E + S–Ent A/P preventive application for 2 weeks (between 0–14 days) before SE P3Tr2a strain addition for 1 week (between 14–21 days); C–control group (without additives); Data are expressed as means and standard deviations (SD). ^a^, ^b^, ^c^, ^d^ Means within lines with different superscript letters are significantly different (*p* < 0.05) using Bonferroni post hoc test; ^A^, ^B^, ^C^, ^D^ Means within lines with different superscript letters are significantly different (*p* < 0.05) using Fisher’s LSD post hoc test. The bold letters mean significant changes.

**Table 3 animals-12-01108-t003:** Bacterial counts (log 10 CFU/g ± SD) in feces of rabbits.

Bacteria	Day	E	S	E + S	C		*p*-Value	
						**Time**	**Treatment**	**Time × Treatment**
*Enterococcus* spp.	0	3.49 ± 0.30	3.49 ± 0.30	3.49 ± 0.30	3.49 ± 0.30			
	14	2.77 ± 0.86	2.51 ± 0.78	2.97 ± 0.38	2.43 ± 1.02	**<0.0001**	**<0.0001**	**<0.0001**
	21	3.09 ± 0.41 ^Aa^	2.52 ± 0.49 ^Ab^	1.46 ± 0.69 ^Bc^	2.76 ± 0.53 ^Aa^			
Coliforms	0	3.59 ± 1.73	3.59 ± 1.73	3.59 ± 1.73	3.59 ± 1.73			
	14	1.17 ± 0.37 ^Aa^	3.61 ± 0.77 ^Bb^	1.60 ± 0.90 ^Aa^	1.67 ± 0.75 ^Aa^	**<0.0001**	**<0.0001**	**<0.0001**
	21	2.26 ± 1.16 ^Aa^	2.68 ± 0.75 ^Aa^	1.00 ± 0.00 ^Bb^	2.14 ± 0.58 ^ABa^			
Amylolytic streptococci	0	3.94 ± 0.60	3.94 ± 0.60	3.94 ± 0.60	3.94 ± 0.60			
	14	3.59 ± 0.12	3.87 ± 0.53	3.22 ± 0.45	3.49 ± 0.30	0.4952	0.9087	0.9973
	21	3.76 ± 1.06	3.48 ± 0.62	3.16 ± 0.57	3.01 ± 0.74			
*Staphylococcus* spp.	0	3.87 ± 0.17	3.87 ± 0.17	3.87 ± 0.17	3.87 ± 0.17			
	14	3.65 ± 0.27	3.70 ± 0.26	3.55 ± 0.29	3.49 ± 0.30	**<0.0001**	**<0.0256**	**<0.0001**
	21	3.30 ± 0.47 ^Aa^	3.02 ± 0.36 ^Bb^	3.67 ± 0.65 ^Cc^	3.60 ± 0.44 ^ACc^			
	0	2.38 ± 0.22	2.38 ± 0.22	2.38 ± 0.22	2.38 ± 0.22			
Methicillin-resistant staphylococci	14	3.65 ± 0.27	3.70 ± 0.26	3.55 ± 0.27	3.48 ± 0.38	**<0.0001**	**<0.0001**	**<0.0001**
	21	4.01± 0.09 ^Aa^	3.55 ± 0.34 ^Bb^	3.06 ± 0.47 ^Cc^	3.79 ± 0.18 ^ABd^			
Total bacteria	0	4.96 ± 0.45	4.96 ± 0.45	4.96 ± 0.45	4.96 ± 0.45			
	14	3.09 ± 0.23 ^a^	3.59 ± 0.71 ^b^	3.45 ± 0.23 ^bc^	3.13 ± 0.44 ^ac^	**<0.0001**	**<0.0001**	**<0.0001**
	21	3.92 ± 0.57 ^Aa^	4.37 ± 1.04 ^Bb^	3.38 ± 0.09 ^Cc^	3.65 ± 0.13 ^ACac^			
SE P3/Tr2a strain	data	NT	1.66 ± 1.20	1.20 ± 0.68	NT		0.3615	

E–Ent A/P application between 0–14 days; S–SE P3/Tr2a application between 14–21 days; E + S–Ent A/P preventive application for 2 weeks (between 0–14 days) before SE P3Tr2a strain addition for 1 week (between 14–21 days); C–control group (without additives); Data are expressed as means and standard deviations (SD). ^a^, ^b^, ^c^, ^d^ Means within lines with different superscript letters are significantly different (*p* < 0.05) using Bonferroni post hoc test; ^A^, ^B^, ^C^ Means within lines with different superscript letters are significantly different (*p* < 0.05) using Fisher’s LSD post hoc test. The bold letters mean significant changes.

**Table 4 animals-12-01108-t004:** Bacterial counts (log 10 CFU/g ± SD) in cecum and appendix of rabbits.

Bacteria	Source	E	S	E + S	C	*p*-Value
*Enterococcus* spp.	cecum	1.63 ± 0.77 ^a^	0.90 ± 0.00 ^b^	0.90 ± 0.00 ^b^	1.56 ± 0.84 ^a^	**<0.0001**
	appendix	0.90 ± 0.00	0.90 ± 0.00	0.90 ± 0.00	1.43 ± 0.41	0.0391
Coliforms	cecum	0.90 ± 0.00 ^a^	1.00 ± 0.17 ^a^	0.90 ± 0.00 ^a^	2.30 ± 0.67 ^b^	**<0.0001**
	appendix	1.43 ± 0.91	1.39 ± 0.63	1.36 ± 0.49	1.05 ± 0.25	0.6810
Amylolytic streptococci	cecum	2.71 ± 0.44	3.35 ± 0.92	3.50 ± 0.67	2.93 ± 0.20	0.1091
	appendix	2.96 ± 1.20	3.13 ± 0.94	3.18 ± 0.66	3.83 ± 0.70	0.2484
*Staphylococcus* spp.	cecum	3.02 ± 0.29	3.64 ± 0.36	3.29 ± 0.88	3.52 ± 0.45	0.1972
	appendix	3.32 ± 0.20	3.65 ± 0.23	3.45 ± 0.16	3.17 ± 0.37	0.0952
Methicillin-resistant staphylococci	cecum	4.03 ± 0.50 ^ab^	4.17 ± 0.19 ^ab^	4.86 ± 1.77 ^a^	3.40 ± 0.27 ^b^	**0.0489**
	appendix	3.40 ± 0.17	3.39 ± 0.23	3.52 ± 0.16	3.75 ± 0.29	0.2088
Total bacteria	cecum	3.22 ± 0.23 ^a^	3.73 ± 0.52 ^ab^	3.91 ± 0.35 ^b^	3.31 ± 0.20 ^a^	**0.0188**
	appendix	3.73 ± 0.77	3.33 ± 0.25	3.50 ± 0.13	3.88 ± 1.22	0.1915
SE P3/Tr2a strain	cecum	NT	0.90 ± 0.00	1.00 ± 0.17	NT	0.9885
	appendix	NT	1.00 ± 0.17	0.90 ± 0.00	NT	0.9885

E–Ent A/P application between 0–14 days; S–SE P3/Tr2a application between 14–21 days; E + S–Ent A/P preventive application for 2 weeks (between 0–14 days) before SE P3Tr2a strain addition for 1 week (between 14–21 days); C–control group (without additives); Data are expressed as means and standard deviations (SD). ^a^, ^b^ Means within lines with different superscript letters are significantly different (*p* < 0.05) using Tukey’s post hoc test. The bold letters mean significant changes.

## Data Availability

Data available upon reasonable request to the corresponding author.

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
