# Peer review of "Preventive Potential of Dipeptide Enterocin A/P on Rabbit Health and Its Effect on Growth, Microbiota, and Immune Response"

_animals, 2022, doi:10.3390/ani12091108_

Round 1
Reviewer 1 Report
A very interesting study was conducted in which the revascularization effect of enterocin dipeptide (Ent) A/P against a methicillin-resistant Staphylococcus epidermidis SE P3/Tr2a strain was evaluated in a rabbit model.
- Please complete the following information after reviewing the manuscript:
- Please describe the handling of samples for microbiological analysis.
-Please describe the handling of blood, centrifugation process, number of rotations, apparatus used for IgA analysis, reading wavelength, etc.
Lines 268-269 should be extended to other studies with probiotics, e.g. with fermented rapeseed meal used in rabbit feed (https://doi.org/10.3390/ani11030716).
The use of fermented feed products in the diet can diversify protein sources and stimulate the beneficial microbiota of the gastrointestinal tract of animals. Similarly, there are beneficial effects of bacteriocins. These are particularly important as they can help to reduce the use of antibiotics for therapeutic purposes through nutritional prophylaxis and optimal development of the immune system.
Author Response
Comments and Suggestions for Authors 1
Thanks for the review of our manuscript and for useful comments according to which the manuscript was revised. The changes in the manuscript are highlighted by colored (red) text.
A very interesting study was conducted in which the revascularization effect of enterocin dipeptide (Ent) A/P against a methicillin-resistant Staphylococcus epidermidis SE P3/Tr2a strain was evaluated in a rabbit model.
- Please complete the following information after reviewing the manuscript:
- Please describe the handling of samples for microbiological analysis.
Freshly voided faeces were collected using 5 nets/1 group.Becuase they were 2 animals in one cage, there was not possible to collect individual faeces samples from the net, which was mounted under cages and also, there was some faeces from the neighboring cage. That was the reason, why we use mixed samples (5 samples from 1 net = 5 samples /1 group). These corrections were also added to the section Materials and methods, see L136-138.
Samples from caecum and appendix (n=8) were collected individually after rabbits slaughtering.
-Please describe the handling of blood, centrifugation process, number of rotations, apparatus used for IgA analysis, reading wavelength, etc.
After blood samples collection from marginal ear veins (n=8), individual samples from 8 rabbits, blood in was incubated for 1 hour at 37 °C in heparinized Eppendorf tubes containing microsferic hydrophilic (MSH) particles (50 µL of MSH particle suspension (ARTIM, Prague, Czech Republic) was mixed with 100 µL of blood in an Eppendorf-type test tube). After incubation, blood smears were prepared and stained in accordance with May-Grünwald and Giemsa-Romanowski. - The materials and methods section was corrected, the details were added to this section, see L 120-125.
IgA analysis - Intestinal wall samples were prepared using the method described by Nikawa et al. For quantitative measurement of immunoglobulin A (IgA) in the intestinal wall, the competitive inhibition enzyme immunoassay technique was used (Rabbit Immunoglobulin A, IgA ELISA kit, Cusabio, Houston, TX, USA). Samples were analyzed using the Multireader Synergy HTX (Biotek USA), at the wavelength 450 nm according to the manufacturer of Cusabio kit. (L131-134).
Lines 268-269 should be extended to other studies with probiotics, e.g. with fermented rapeseed meal used in rabbit feed (https://doi.org/10.3390/ani11030716). The use of fermented feed products in the diet can diversify protein sources and stimulate the beneficial microbiota of the gastrointestinal tract of animals. Similarly, there are beneficial effects of bacteriocins. These are particularly important as they can help to reduce the use of antibiotics for therapeutic purposes through nutritional prophylaxis and optimal development of the immune system.
Thank you for this usefull comment, the reference was added to this section (L267).
Reviewer 2 Report
The manuscript by Simonová et al. represents an approach to study the protective impact of dipeptide enterocin A/P against methicillin-resistant Staphylococcus epidermidis SE P3/Tr2a strain in growing rabbits. The subject itself is surely worthy of investigation. However, some points need to be addressed as follows:
- The hypothesis of the study should be clarified before the aim at the end of the Introduction section.
- Line 92, delete " Slaughtering and Sampling".
- Lines 96-97: Remove the data from the text as it is already displayed in Table 2. Just add the overall mean ± SD.
- Were the rabbits adapted to the experimental conditions before the start of the experiment, and for how long?
- Why was it chosen to add the tested additive to drinking water instead of feed? Is it difficult to add to the feed?
- Since the treatments time are different, I suggest adding a figure to illustrate the experimental design.
- Methods for estimating the nutrients in Table 1 were not found in the M&M section.
- Is there any mortality throughout the experiment?
- Table 2 shows significant differences in the initial weight, and thus, the differences at 14 and 21 d of the experiment cannot be owned to the treatments only. Therefore, the initial weight must be taken as a covariable during the statistical analysis.
Author Response
Comments and Suggestions for Authors
Thanks for the review of our manuscript and for useful comments according to which the manuscript was revised. The changes in the manuscript are highlighted by colored (red) text.
The manuscript by Simonová et al. represents an approach to study the protective impact of dipeptide enterocin A/P against methicillin-resistant Staphylococcus epidermidis SE P3/Tr2a strain in growing rabbits. The subject itself is surely worthy of investigation. However, some points need to be addressed as follows:
- The hypothesis of the study should be clarified before the aim at the end of the Introduction section.
The hypothesis of the study is included before the aims, and it was corrected see: ..“but the prevention of pathogenic and resistant bacteria multiplication using bacteriocins as feed additives is also important.“ (L68-69).
- Line 92, delete " Slaughtering and Sampling".
It was deleted.
- Lines 96-97: Remove the data from the text as it is already displayed in Table 2. Just add the overall mean ± SD.
It was corrected for 1091.8 g ± 149.3 (Table 2)., see L 86.
- Were the rabbits adapted to the experimental conditions before the start of the experiment, and for how long?
There was no specific adaptation period before the experiment, ususally we use 35 days aged rabbits, right after weaning to demonstrate daily conditions at rabbit farms.
- Why was it chosen to add the tested additive to drinking water instead of feed? Is it difficult to add to the feed?
Regarding the application form of tested additives, all additives are tested before their application to animals under laboratory conditions, Usually, their stability, solubility, storage conditions, etc. are tested. Concerning the beneficial bacteria and their products, bacteriocins, it is easier to applied them to the drinking water. We have several experiments where bacterial strains were added to the feed, of course, they were firstly lyophilized, added to pellets and resolved and tested to their viability, because of high temperature during pelletization – on the basis of these results, selected strains were added to probiotic preparation Prorabbit (Inproco, Slovakia; it is probiotic powder for rabbits and rodents). Our selected strains for testing their application form did not show significant differences in their antimicrobial or another beneficial properties even there were applied as fresh culture to water or lyophilized to feed.
- Since the treatments time are different, I suggest adding a figure to illustrate the experimental design.
The figure was added (L108-112).
- Methods for estimating the nutrients in Table 1 were not found in the M&M section.
This section was improved, the data accroding nutrients analyses were added, see: The chemical analyses of feed ingredients were conducted according to AOAC [19] and Van Soest et al., [20], L92-93.
- Is there any mortality throughout the experiment?
The data regarding the mortality were added to the section of Results, Table 2
- Table 2 shows significant differences in the initial weight, and thus, the differences at 14 and 21 d of the experiment cannot be owned to the treatments only. Therefore, the initial weight must be taken as a covariable during the statistical analysis.
I agree with the Reviewer that differences at days 14 and 21 are not related only the treatment, but outgoing from results of FCR – lower FCR during enterocin A/P application, we hypothesized beneficial effect of Ent A/P on FCR and growth performance. The sections of Results and Discussion were improved, regarding also growth and FCR.
Reviewer 3 Report
The authors investigated the preventive potential of dipeptide enterocin A/P on rabbit health and its effect on growth, microbiota, and immune response. They designed four treatments to test their hypothesis. This manuscript (MS) was clearly written and easy to understand. This work can help the sustainability of this species farming. However, some major issues significantly compromised the quality of this MS.
Major comments:
- First, the manuscript needs to be edited by a native English speaker to improve the language of the MS and fix errors.
- The statistical analysis is required to be updated. Please see the comment in the “minor comment section”.
However, I have touched on some more points that can contribute to the improvement of this MS.
Minor comments
- Line 17, what is A/P?
- Line 117-120, please explain why firstly you water them 100 ml and then ad libitum.
- Line 146, please revise it.
- Statistical treatments are required to be updated. When the interaction is significant, you should unpack the treatments and compare them. When the interaction is not significant, you should compare pooled data. Please find the below article to understand the process. https://onlinelibrary.wiley.com/doi/10.1111/jpn.13659
- Please update the result and discussion with the above change.
- Line 264-268, growth is a phenotype different from Please revise this sentence. The possible animal has good growth but not FCR and vice versa.
- Line 276, can enhance
- The discussion section was written very well. I do not have any comments. But a revision is required to improve the language of the MS.
Best regards
Author Response
Comments and Suggestions for Authors
Thanks for the review of our manuscript and for useful comments according to which the manuscript was revised. The changes in the manuscript are highlighted by colored (red) text.
The authors investigated the preventive potential of dipeptide enterocin A/P on rabbit health and its effect on growth, microbiota, and immune response. They designed four treatments to test their hypothesis. This manuscript (MS) was clearly written and easy to understand. This work can help the sustainability of this species farming. However, some major issues significantly compromised the quality of this MS.
Major comments:
- First, the manuscript needs to be edited by a native English speaker to improve the language of the MS and fix errors
Errors were checked and corrected, the English language was corrected.
- The statistical analysis is required to be updated. Please see the comment in the “minor comment section”.
The statistical analysis was checked and updated.
However, I have touched on some more points that can contribute to the improvement of this MS.
Minor comments
- Line 17, what is A/P?
Enterocin A (entA) is a class 2a bacteriocin isolated from E. faecium CTC492 and shows inhibition of E. faecalis, Pediococcus, Listeria, and some strains of E. faecium (Aymerich et al., 1996). Enterocin P is an enterocin produced by E. faecium P13 (Cintas et al., 1997).
Our tested enterocin, previously named as enterocin EK13, produced by the Enterococus faecium EK13 strain – included in the section Introduction, L 75-78); this enterocin EK13 was purified to its homogenity and characterized as enterocin A and also as enterocin P and it has genes for encoding of enterocins A and P; for this reason it is called that dipeptide enterocin A/P.
- Line 117-120, please explain why firstly you water them 100 ml and then ad libitum.
To ensure that rabbits received the applied volume of tested additives (L104).
- Line 146, please revise it
I apologize, but I am not really sure, what can I revise?
- Statistical treatments are required to be updated. When the interaction is significant, you should unpack the treatments and compare them. When the interaction is not significant, you should compare pooled data. Please find the below article to understand the process. https://onlinelibrary.wiley.com/doi/10.1111/jpn.13659
The statistical analysis was checked.
- Please update the result and discussion with the above change.
The results were checked and corrected.
- Line 264-268, growth is a phenotype different from Please revise this sentence. The possible animal has good growth but not FCR and vice versa.
I agree with the Reviewer, that animals in all experimental groups had better weight gain that in control group (these data respectively findings are included in the sections Results and Discussion. But more important is the lower FCR compared to control respectively compared between E and S groups – the body weight was higher (not significantly), but the FCR value in E group was lower by more than 2 tenths, which is a good finding from the economic point of view – less feed = less money for better/faster growth. This section was corrected/improved, see L 266 :...”this finding is important from the economic point of view.“
- Line 276, can enhance
It was corrected.
- The discussion section was written very well. I do not have any comments. But a revision is required to improve the language of the MS.
The English language was checked.
Round 2
Reviewer 2 Report
No further comments to be addressed
Reviewer 3 Report
Unfortunately, the authors did not improve the quality of the MS. I gave them even an article as an example of how to analyse data and report them but they did not notice and check that out.
Kind regards